# ToolACE-MT: Non-Autoregressive Generation for Agentic Multi-Turn Interaction

**Xingshan Zeng**[1], **Weiwen Liu**[2]\*, **Lingzhi Wang**[3], **Liangyou Li**[1], **Fei Mi**[1], **Yasheng Wang**,
**Lifeng Shang**[1], **Xin Jiang**[1], **Qun Liu**[1]

[1]Huawei Technologies Co., Ltd
[2]Shanghai Jiao Tong University
[3]Harbin Institute of Technology, Shenzhen
`zeng.xingshan@huawei.com, wwliu@sjtu.edu.cn`

## ABSTRACT

Agentic task-solving with Large Language Models (LLMs) requires multi-turn, multi-step interactions, often involving complex function calls and dynamic user-agent exchanges. Existing simulation-based data generation methods for such scenarios rely heavily on costly autoregressive interactions between multiple LLM agents, thereby compromising the practical efficiency of agentic data generation. In this paper, we propose ToolACE-MT, a novel Non-Autoregressive Iterative Generation framework for constructing high-quality multi-turn agentic dialogues. ToolACE-MT generates full conversational trajectories through three stages: coarse-grained initialization, iterative refinement, and offline verification. The initialization phase builds a structurally complete yet semantically coarse dialogue skeleton; the iterative refinement phase introduces realistic complexities and continued refinement via mask-and-fill operations; and the offline verification phase ensures correctness and coherence via rule- and model-based checks. Experiments demonstrate that ToolACE-MT enables efficient, effective and generalizable agentic data generation, offering a new paradigm for high-quality data construction in tool-augmented LLM scenarios.

## 1 INTRODUCTION

Large Language Models (LLMs) have demonstrated remarkable abilities in open-ended generation, reasoning, and instruction following Guo et al. (2025); Wang et al. (2024); Jiang et al. (2024). Beyond passive language understanding, a growing frontier in LLM research involves agentic task-solving, where models take on the role of autonomous agents interacting with users and environments over multi-turn dialogues Wang et al. (2023); Luo et al. (2025). These settings often involve multiple function calling[1] and adaptive decision-making, significantly broadening the applicability of LLMs in real-world scenarios.

To enable such agentic capabilities, high-quality multi-turn multi-step interaction data is essential. Multi-turn refers to multiple exchanges between the user and the assistant, while multi-step denotes task completion that requires executing a sequence of dependent actions, often through function calls. Together, they reflect the complexity of real-world agentic scenarios where task states are partially observable. However, constructing such data is inherently challenging: it requires generating complex but solvable tasks, maintaining coherent user-agent exchanges, and accurately simulating tool behaviors. A promising direction lies in multi-agent simulations, where multiple LLMs are assigned roles including user, assistant and tool to collaboratively generate full conversations through autoregressive interactions Liu et al. (2025); Prabhakar et al. (2025). While effective at generating natural conversations, these approaches have several drawbacks: 1) They are computationally costly due to extended back-and-forth interactions, where each new turn must be generated one-by-one based on all previous context; 2) Task complexity and dialogue length are implicitly determined by

---

\*Corresponding Author.

[1]In this paper, function calling, tool calling and tool use are used interchangeably.

model interactions and are difficult to constrain explicitly, which poses challenges for fine-grained data design; 3) Most critically, since assistant behaviors are generated autoregressively without access to global context, i.e. the overall task and dependencies between steps, it is difficult for the assistant to optimize the overall output structure and ensure consistency at each step. This lack of holistic awareness hinders factual accuracy, tool-use consistency, and task solvability, especially in scenarios requiring long-term planning. As a result, the quality of generated data largely depends on the capability of the LLMs playing the assistant role, often resembling a form of knowledge distillation from larger (assistant) models.

In this paper, we propose ToolACE-MT, a novel framework for constructing multi-turn dialogues involving agentic tool-use behaviors, inspired by Non-Autoregressive Translation (NAT) and masked diffusion language models Gu et al. (2018); Sahoo et al. (2024), which have been shown to be more efficient in language generation. Unlike traditional autoregressive multi-agent simulations (MAS), ToolACE-MT generates full conversational trajectories through a non-autoregressive pipeline consisting of three stages (see Figure 1): 1) Coarse-Grained Initialization: A coarse but structurally complete dialogue skeleton is generated, specifying overall tasks and action trajectory. 2) Iterative Refinement: Through carefully designed mask-and-fill procedures, the skeleton is progressively enriched with complexity injection and improved with reasonability refinement. 3) Offline Verification: Rule- and model-based checks are applied, filtering out invalid samples.

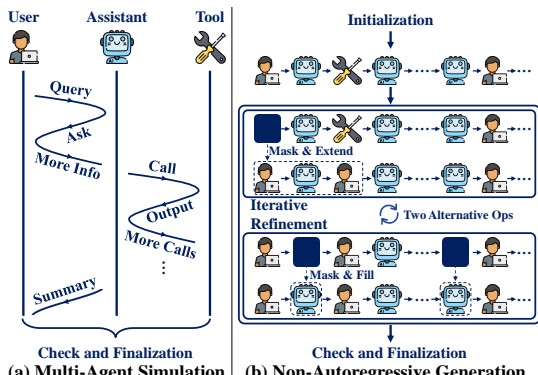

Figure 1: Multi-Agent Simulation v.s. our proposed Non-Autoregressive Generation.

ToolACE-MT offers notable improvements in generation efficiency and complexity control, while preserving functions to generate high-quality agentic data. Through the iterative refinement operations, ToolACE-MT also provides flexible scaling, enabling budget-constrained data generation.

Exerimental results on several agentic multi-turn benchmarks, including BFCL-v3 (Berkeley Function Calling Leaderboard) Yan et al. (2024), $\tau$-Bench Yao et al. (2025) and ACEBench Chen et al. (2025), show that models trained with ToolACE-MT generated data outperform those with autoregressive MAS. Data analysis and ablation studies confirm the efficiency and effectiveness of our generation pipeline, and more experiments show the generalizability to different backbones.

In summary, our contributions are as follows:

- We propose ToolACE-MT, a non-autoregressive iterative generation framework for agentic multi-turn interaction.
- The iterative refinement strategy in ToolACE-MT enables flexible complexity enhancement and quality improvement, which can be further scaled based on budgets.
- We provide extensive empirical evidence showing that ToolACE-MT enables efficient, high-quality generation of agentic dialogue data suitable for tool-use LLM training.

## 2 RELATED WORK

**Agentic Data Synthesis.** LLM agents equipped with external tools have shown realistic problem-solving capabilities Qin et al. (2024); Gou et al. (2024); Lu et al. (2025). As current LLMs still face challenges with complex tasks Mialon et al. (2024); Yao et al. (2025), learning from synthesized agentic data offers a promising direction. Early work focuses on single-turn synthesis, where the agent receives a one-time query and responds accordingly Patil et al. (2023); Zeng et al. (2023); Qin et al. (2024); Liu et al. (2024). However, real-world tasks often involve multi-turn, multi-step interactions, prompting recent studies to explore conversational data synthesis via multi-agent simulations Tang et al. (2023); Liu et al. (2025); Wang et al. (2025). Closest to our approach is Prabhakar et al. (2025), which adopts a two-stage synthesis process. While their first stage resembles

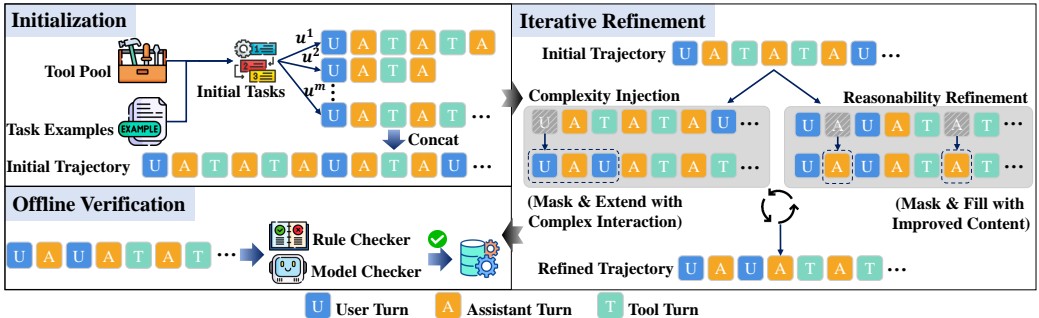

Figure 2: Overall workflow for our ToolACE-MT framework.

ours which generates task configurations and ground-truth answers, they still rely on multi-agent simulations in the second stage to collect full interaction trajectories.

**Agentic Model Training.** Fine-tuning on synthesized data remains central to agentic model training Qin et al. (2024); Liu et al. (2025); Prabhakar et al. (2025). With reinforcement learning (RL) proving effective in enhancing LLM reasoning Shao et al. (2024); Guo et al. (2025), agentic RL has emerged as a promising alternative for developing agentic capabilities in recent research Feng et al. (2025); Qian et al. (2025); Zhang et al. (2025); Jin et al. (2025). While agentic RL may reduce the complexity of data synthesis by enabling learning from sparse or indirect supervision, high-quality synthesized data remains essential to guide and stabilize training.

**Non-Autoregressive Generation.** To overcome the inefficiencies and quality limitations in certain scenarios of traditional autoregressive generation, where tokens are produced one-by-one, non-autoregressive approaches have been proposed Gu et al. (2018); Xiao et al. (2023). These methods include CTC-based objectives Libovický & Helcl (2018), iterative refinement with Mask-Predict Ghazvininejad et al. (2019), and insertion and deletion strategies Gu et al. (2019). Inspired by this line of work, we extend the non-autoregressive paradigm to the turn level, enabling more efficient and coherent agentic dialogue data synthesis.

## 3 METHOD

### 3.1 PROBLEM FORMULATION

Solving complex agentic tasks requires multi-turn interaction between the AI assistant and the user/environment. During this process, the assistant may ask clarification questions or interact with the environment to accomplish the user's tasks. The interaction with the environment can also be multi-step, involving multiple function calls either at a single turn (called parallel function calls) or one after another (dependent function calls). This task-solving process can be formulated as a partially observable Markov decision process (POMDP), defined as $(\mathcal{S}, \mathcal{U}, \mathcal{A}, \mathcal{O}, \mathcal{T}, \mathcal{R})$, where $\mathcal{S}$ is the state space, $\mathcal{U}$ is the task space, $\mathcal{A}$ is the action space, $\mathcal{O}$ is the observation space, $\mathcal{T} : \mathcal{S} \times \mathcal{A} \to \mathcal{S} \times \mathcal{O}$ is the transition function, and $\mathcal{R}$ is the reward function evaluating the overall process.

We define one interaction between the assistant and the user/environment as a single turn. The assistant executes a sequence of actions $(a^1, a^2, \cdots, a^n)$, where each $a^t \in \mathcal{A}$, to accomplish the user's tasks $(u^1, u^2, \cdots, u^m)$, where each $u^t \in \mathcal{U}$. A single conversation may involve multiple tasks issued incrementally. Each action $a^t$ can be either a function call or a natural language response to the user. The corresponding observation $o^t$ is either the tool's output or the user's follow-up message. Importantly, the environment state $s^t$ after executing $a^t$ remains latent to both the assistant and the user. The interaction concludes when all user tasks $u^t$ are completed or the maximum number of turns is reached. The final reward $r \in \mathcal{R}$ is computed based on the cumulative state changes and, optionally, the action sequence, depending on the level of granularity desired.

This interaction results in a sequence of alternating observations and actions as a trajectory, $C = (o^0, a^1, o^1, \cdots, o^{n-1}, a^n)$, where $o^0$ is the user's initial message and $o^t$ is the tool output or user reply following $a^t$. $C$ constitutes the target for data generation, i.e., multi-turn conversational data.

## 3.2 OVERVIEW

To construct high-quality conversational sequences $C$, prior work applies multi-agent simulations Liu et al. (2025); Prabhakar et al. (2025), where LLMs simulate the user (producing observations after assistant responses), the tools (producing outputs after function calls, can be either actual tools or simulated ones), and the assistant (producing actions). While this approach is shown to work, it is costly, hard to verify and complexity control. We propose a more efficient and controllable method: Non-Autoregressive Iterative Generation (ToolACE-MT), inspired by non-autoregressive translation (NAT) and diffusion models Gu et al. (2018); Sahoo et al. (2024).

Our pipeline consists of three stages: 1) Initialization 2) Iterative Refinement, and 3) Offline Verification. Figure 2 illustrates the overall workflow. We describe each stage in detail below.

## 3.3 COARSE-GRAINED INITIALIZATION

The goal of the initialization stage is to generate a coarse but structurally complete skeleton of a multi-turn conversation. Both the user tasks and the conversational trajectory are initialized in a loosely coupled fashion, enabling later stages to enhance coherence and inject complexity. This stage lays the groundwork for efficient and controllable non-autoregressive generation.

### 3.3.1 TASK INITIALIZATION.

We begin by sampling or specifying a candidate tool list from a predefined tool pool Liu et al. (2025); Wang et al. (2025). The overall task is then generated with the following components: 1) a set of subtasks $(u^1, u^2, \cdots, u^m)$ (with $m$ predefined per instance), 2) the required tools for each subtask, 3) and the number of steps required for tool usage for each subtask.

This step serves as high-level planning, outlining the overall trajectory without finalizing all the details. To ensure coverage across domains and promote task diversity, we curate both actual and simulated tools from prior work Qin et al. (2024); Liu et al. (2025) and handcraft initial task examples. The examples will be further enriched during the data generation process.

### 3.3.2 TRAJECTORY INITIALIZATION.

Given the tool list and the generated user tasks, we generate an initial conversational trajectory skeleton $C = (o^0, a^1, o^1, \cdots, a^n)$ by composing subtask trajectories sequentially. For each subtask $u^t$, we generate a sub-trajectory $C_t = (o_t^0, a_t^1, \cdots, a_t^s, \cdots)$ based on the generated subtask metadata (i.e., tool requirements and number of steps) and previously generated sub-trajectories $(C_0, \ldots, C_{t-1})$. The final initial trajectory is obtained by concatenating all subtask trajectories: $C = C_0 \cup C_1 \cup \cdots \cup C_m$.

Notably, each sub-trajectory is generated with tool calls and outputs generating in parallel to ensure consistency. Also, to simplify downstream refinement, we enforce that the user query $o_t^0$ contains all necessary information (e.g., parameter values for function calls), and all subsequent observations $o_t^s$ ($s \neq 0$) are tool outputs.

This structure ensures proper alternation between action types (e.g., function calls follow by tool outputs, and natural language responses follow by user replies) and facilitates easier post-processing. Note that this stage prioritizes structural completeness over semantic correctness. The generated content may be shallow or partially inconsistent and need to be refined later.

## 3.4 ITERATIVE REFINEMENT

In this stage, we enhance the initial trajectory through multiple refinement passes, improving both complexity and semantic coherence. Inspired by Masked-Predict Ghazvininejad et al. (2019), we iteratively apply mask-and-extend/fill to progressively improve the trajectory $C$ (see Figure 3).

### 3.4.1 COMPLEXITY INJECTION.

To better simulate real-world dialogues, we inject complexity into the initialized trajectories. The injection types include:

Figure 3: Illustration figure for Iterative Refinement process.

- *Clarification*: user gives incomplete information following by assistant's clarification question.

- *Tool awareness*: assistant recognizes unsupported tasks and user updates the tool list.

- *Error simulation*: tool call fails, resulting assistant reflecting and adjusting actions.

- *Non-function-calling requirements*: e.g., chitchat or open-ended user inputs to increase diversity.

Each kind of injection is implemented via a specific mask-and-extend operation. The "mask" operation refers to replacing the whole content of one turn with a placeholder, and "extend" means to fill with revised content and add additional turns. For instance, if masking at turn $o^t$, we generate: $(o^0, a^1, \cdots, a^t, o', a', o'', a^{t+1}, \cdots, a^n) = f_{\text{LLM}}(\sigma, (o^0, a^1, \cdots, a^t, X, a^{t+1}, \cdots, a^n))$, where $\sigma$ is the selected injection type and $X$ indicates the masked turn.

Since the trajectory is clean by initialization, we can easily maintain an injection log to record which turns have been modified and avoid redundant modifications.

### 3.4.2 REASONABILITY REFINEMENT.

Apart from injecting complexity, we perform another refinement pass to enhance logical consistency and coherence. This includes checking whether tool calls have appropriate parameters, ensuring natural language responses are contextually relevant, verifying dialogue flow and resolving inconsistencies.

We adopt a mask-and-fill strategy that randomly masks several non-adjacent turns and regenerates them using an LLM. Initially, all turns have equal selection probability, but each time a turn is chosen, its probability is reduced, encouraging diverse turns to be refined across iterations. To prevent problematic refinement, an LLM-based judger is used to determine whether to adopt the newly generated content or retain the original ones.

For each trajectory, complexity injection and reasonability refinement are both applied alternatively in an iterative manner, until all turns have been refined or the predefine refinement count for each type is reached.

### 3.5 OFFLINE VERIFICATION

Given the extensive use of LLMs in the aforementioned stages, hallucination remains a critical issue, particularly in long multi-turn dialogues involving large tool lists Liu et al. (2023). To address this, we conduct offline verification on the refined trajectories using a hybrid approach that combines rule-based and model-based methods Liu et al. (2024; 2025).

For rule-based, we evaluate several aspects, including dialogue and tool-calling format compliance, executability (when real tools are available), repetition, and identifiable hallucinations that can be detected with rules, such as references to special IDs that do not appear in the history.

For model-based, inspired by Liu et al. (2025), we decompose the evaluation into multiple sub-questions. Each sub-question is handled independently by an LLM-based checking expert, ensuring modular and focused assessment. The final decision is made based on the aggregation of the individual outputs. We focus on semantic coherence and the detection of complex hallucinations that rule-based methods may miss in this step.

# 4 EXPERIMENTS

## 4.1 EXPERIMENTAL SETUP

**Dataset Construction.** We construct in total 8000 training instances using ToolACE-MT for experiments. For comparison, we also construct 8000 instances with multi-agent simulation (MAS) method introduced in Wang et al. (2025). For fair comparison, we leverage the same candidate tool pool and LLM (GPT-4o-2024-11-20) for generation, the same offline verification is also applied.

For each instance, the number of subtasks is sampled from $[2, 5]$, and each subtask contains $[1, 6]$ steps. During iterative refinement, we randomly inject 1 to 3 different types of complexity to avoid redundant patterns (such as repeatedly asking clarification questions for the same subtask) which could harm dialogue naturalness. Each instance undergoes reasonability refinement up to 5 times (More refinement can be applied, while this is empirically cost-effective balance).

**Models.** We use LLaMA3.1-8B-Instruct AI@Meta (2024) as the base model in our main experiments. Other models, including Qwen2.5-Instruct-series Yang et al. (2024) (0.5B, 1.5B, 3B and 7B) and Qwen3-8B Yang et al. (2025), are also tested to validate the generalizability of our method. To verify the effectiveness of our proposed three stages, we also train models with data without offline verification and iterative refinement for ablation study.

**Benchmarks and Evaluation.** We conduct experiments on several representative benchmarks targeting on the multi-turn capabilities of tool-augmented LLMs, including the Berkeley Function Call Leaderboard (BFCL-v3) Yan et al. (2024), $\tau$-Bench Yao et al. (2025), and ACEBench Chen et al. (2025). As we focus on the realistic multi-turn capabilities, we mainly present and analyze results on the categories related to Multi-Turn categories. Results in single turn are also listed (for BFCL, while those for ACEBench listed in Appendix Table 7) to show the robustness.

**Training Details.** Given resource constraints, we employ the parameter-efficient fine-tuning method LoRA Hu et al. (2022) for model training. All model modules are configured for LoRA fine-tuning, with a rank of 16 and an alpha value of 32. Training is performed with a global batch size of 64 and a learning rate of $1 \times 10^{-4}$, following a cosine learning rate schedule with a warmup ratio of 0.1. Please refer to Appendix Table 6 for hyper-parameter tuning details.

Table 1: Data Statistics and corresponding quality assessment comparing MAS and ToolACE-MT. "D" is short for "Dialogue", "UT" is short for "User Turn", and "CT" is short for "Call Turn". "SS" (Semantic Similarity) and "EnR" (Entailment Rate) evaluate coherence, and "H" (Entropy) and "Dist-3" (Distinct-3) assess diversity.

| Method | Statistics | | | | | Quality | | | |
|---|---|---|---|---|---|---|---|---|---|
| | Tool # /D | Turn #/D | UT #/D | CT #/D | Call #/UT | SS | EnR | H | Dist-3 |
| **MAS** | 8.8 | 28.0 | 5.8 | 8.7 | 2.3 | 65.23 | 43.60 | 7.92 | 0.319 |
| **ToolACE-MT** | 8.4 | 23.7 | 3.4 | 8.5 | 3.7 | 68.34 | 50.71 | 9.28 | 0.357 |

## 4.2 DATA QUALITY ASSESSMENT

For clarity, Table 1 reports the core data statistics and quality assessment results comparing MAS and ToolACE-MT. In terms of data statistics, ToolACE-MT produces fewer user turns per dialogue but more tool calls per turn. This pattern indicates that we achieve tasks with fewer conversational steps while maintaining a stronger focus on tool invocation, reflecting improved efficiency in multi-step task completion. For quality assessment, following Wang et al. (2025), we evaluate coherence and diversity. Coherence measures the logical consistency and contextual alignment across turns, where we calculate the rate of entailment relation (EnR) and semantic similarity (SS) between two consecutive turns. Diversity captures the variety of linguistic expressions and interaction patterns, where Shannon entropy (H) based on the word frequency and Distinct-N Score with N = 3 are reported (please refer to Wang et al. (2025) for details). The results show that our method yields dialogues with higher coherence and greater diversity.

## 4.3 Main Results

### 4.3.1 Results on BFCL

Table 2: Accuracy comparison (%) on BFCL-v3 (Last updated on 2025-08-26). The table is divided into three parts: Proprietary Models, Open-Source Models, and our experimental models trained based on Llama3.1-8B-Inst. The best results for the last part in each category are marked in **bold**. The second best results are underlined.

| Models | Multi-Turn | | | | | Single-Turn | | Hallucination | | Overall |
| --- | --- | --- | --- | --- | --- | --- | --- | --- | --- | --- |
| | *Overall* | *Base* | *Miss Func* | *Miss Param* | *Long Context* | *Non-Live* | *Live* | *Rel* | *Irrel* | *Overall* |
| **GPT-4o-2024-11-20** | 50.00 | 61.00 | 45.50 | 35.50 | 58.00 | 86.81 | 78.85 | 83.33 | 81.31 | 71.71 |
| **Gemini-2.5-Pro-Preview-05-06** | 34.62 | 39.50 | 29.50 | 31.50 | 38.00 | 65.35 | 74.59 | 33.33 | 90.67 | 59.94 |
| **DeepSeek-V3-0324** | 29.87 | 41.00 | 21.00 | 23.00 | 34.50 | 88.54 | 77.34 | 83.33 | 76.49 | 64.71 |
| **Llama3.1-70B-Inst** | 12.50 | 17.00 | 13.00 | 10.50 | 9.50 | 89.98 | 62.24 | 100 | 54.78 | 53.57 |
| **Llama3.1-8B-Inst** | 9.25 | 12.00 | 10.00 | 7.00 | 8.00 | 84.21 | 61.08 | 77.78 | 48.82 | 49.57 |
| **Multi-Agent Simulation** | 31.38 | 46.50 | 19.00 | 31.00 | 29.00 | 80.29 | **78.05** | 72.22 | **90.11** | 64.17 |
| **ToolACE-MT** | **40.25** | **57.50** | **31.50** | **34.00** | **38.00** | **84.94** | 71.52 | 77.78 | 72.83 | **65.41** |
|   **- Offline Verification** | 32.50 | 48.00 | 25.50 | 25.50 | 31.00 | 79.71 | 75.52 | **83.33** | 80.65 | 63.01 |
|    **- Iterative Refinement** | 20.88 | 39.00 | 12.00 | 10.50 | 22.00 | 75.92 | 61.57 | 72.22 | 46.25 | 52.10 |

Table 2 shows the experimental results on BFCL-v3. The results demonstrate that ToolACE-MT significantly improves multi-turn function calling accuracy, outperforming strong open-source and even some proprietary models (e.g. Gemini-2.5-Pro). Specifically, ToolACE-MT achieves a $40.25\%$ multi-turn accuracy, a $31\%$ absolute improvement over the base model Llama3.1-8B-Inst ($9.25\%$), and even higher than models with larger sizes like Llama3.1-70B ($12.50\%$) and DeepSeek-V3 ($29.87\%$). Compared to the Multi-Agent Simulation (MAS) ($31.38\%$), ToolACE-MT also achieves consistently better results across all multi-turn subcategories. These findings highlight the effectiveness of our ToolACE-MT framework in constructing coherent, contextually grounded dialogues with accurate tool usage.

Beyond multi-turn performance, ToolACE-MT also demonstrates strong generalization to single-turn and hallucination evaluation settings. It achieves $84.94\%$ accuracy in the non-live single-turn category, on par with the base model Llama3.1-8B-Inst, while MAS fails to preserve ($80.29\%$). An interesting finding is that performance on the live single-turn category achieves less improvement compared to MAS, which we attribute to the nature of real user queries in live category: they are often ambiguous. Models trained with richer multi-turn supervision tend to favor asking clarification questions before executing tool calls for ambiguous queries. This behavior reflects a trade-off between cautious multi-turn planning and aggressive single-turn execution.

Ablation studies further validate the effectiveness of our proposed three-stage framework. Removing the Offline Verification stage results in a $2.4\%$ absolute drop in overall performance, underscoring its importance in filtering out problematic or inconsistent instances. Further removing the Iterative Refinement stage leads to a substantial performance decline across all evaluation categories. Upon manual inspection of the generated initial dialogues, we observe that many are either overly simplistic or contain semantic flaws. This highlights the critical role of Iterative Refinement in improving dialogue coherence and increasing complexity.

Interestingly, the Reasonability Refinement part in Iterative Refinement also provides partial functionality similar to that of Offline Verification, such as identifying and correcting inconsistencies during generation. The complementary relationship between these two stages and their overlapping effects will be further discussed in later subsection.

### 4.3.2 Results on More Benchmarks

We further conduct experiments on ACEBench and $\tau$-Bench, which involve more realistic multi-turn interaction settings. In the Agent category of ACEBench and both the Airline and Retail domains in $\tau$-Bench, an LLM simulates the user to interact with the assistant model. Unlike BFCL-v3, they do

Table 3: Accuracy (%) on Multi-turn (MT) and Agent ("EA": End-to-End Accuracy; "PA": Process Accuracy) categories of ACEBench (En).

| Models | MT | EA | PA |
|---|---|---|---|
| GPT-4o-2024-11-20 | 68.0 | 56.0 | 77.8 |
| Llama3.1-70B-Inst | 61.0 | 41.0 | 62.5 |
| Llama3.1-8B-Inst | 24.0 | 6.7 | 18.3 |
| Multi-Agent Simulation | 48.0 | 6.7 | 15.0 |
| ToolACE-MT | **51.0** | **8.4** | **34.0** |
|   - Offline Verification | 44.0 | 1.7 | 28.5 |
|   - Iterative Refinement | 34.0 | 1.7 | 22.8 |

Table 4: Pass@1 (%) comparison on $\tau$-Bench. Results marked with $*$ indicate anomalies due to known evaluation limitations in $\tau$-Bench.

| Models | Retail | Airline | Avg. |
|---|---|---|---|
| GPT-4o-2024-11-20 | 60.4 | 42.0 | 51.2 |
| Llama3.1-70B-Inst | 50.4 | 26.0 | 38.2 |
| Llama3.1-8B-Inst | 6.1 | 26.0* | 16.1 |
| Multi-Agent Simulation | 21.7 | 10.0 | 15.9 |
| ToolACE-MT | **25.2** | **16.0** | **20.6** |
|   - Offline Verification | 22.6 | 6.0 | 14.3 |
|   - Iterative Refinement | 9.5 | 6.0 | 7.8 |

not provide fixed ground-truth trajectories. Instead, a dialogue is considered successful and rewards are assigned accordingly only if the assistant achieves correct states.

We present the results on ACEBench in Table 3, including results on Multi-Turn and Agent categories. For Agent category, we report both End-to-End Accuracy (EA) and Process Accuracy (PA), where PA assesses the consistency between predicted trajectories and ground-truths. As can be seen, ToolACE-MT outperforms MAS baseline across all three metrics, with particularly strong gains in Agent PA, indicating better planning and execution consistency. Ablation studies further confirm the contributions of the Offline Verification and Iterative Refinement stages, each contributing to performance improvement. Notably, the Agent EA remains low for all 8B-scale models, highlighting the significant challenge this setting poses for smaller LLMs.

The results on $\tau$-Bench (shown in Table 4) show a consistent trend, with ToolACE-MT outperforming MAS baseline and the ablation models. Interestingly, the base model Llama3.1-8B-Inst obtains a higher score of 26% in the Airline domain, surpassing all trained models. This counterintuitive outcome can be attributed to a known evaluation limitation in $\tau$-Bench Zhu et al. (2025): several instances define empty actions as the correct responses, assessing the assistant's ability to recognize unsolvable user requirements. When a model lacks sufficient capability and consistently fails to produce valid function calls, it may coincidentally align with these empty actions and receive positive rewards, despite not demonstrating actual understanding. However, this phenomenon does not persist after training, which ultimately leads to lower evaluation scores.

### 4.4 DATA EFFICIENCY

Table 5: Cost and quality comparisons for the two generation methods. "with GPT-4o/GPT-4o-mini" indicates the used LLMs. "Cost" refers to the corresponding cost (total API call times, input and output tokens and final pricing) for generating 8000 samples, and "Quality" is the overall pass rate (%) when applying Offline Verification. "Performance" is the average accuracy in BFCL-v3.

| Method | Cost | | | | Quality | Performance |
|---|---|---|---|---|---|---|
| | API Call times | Input (#) | Output (#) | Pricing (USD) | | |
| **MAS** with GPT-4o | 275k | 652,145k | 10,712k | 1,737 | 61.1 | 64.17 |
| **ToolACE-MT** with GPT-4o | 188k | 478,115k | 18,480k | 1,380 | 72.3 | 65.41 |
| **ToolACE-MT** with GPT-4o-mini | 394k | 855,391k | 33,062k | 148 | 48.7 | 60.13 |

**Data Generation Efficiency.** We compare the cost and quality of generating agentic dialogue data using ToolACE-MT versus MAS. As shown in Table 5, MAS yields a lower Offline Verification pass rate (61.1% vs. 72.3%), therefore requiring a larger initial dataset and in total 275k API calls, to obtain 8,000 valid samples, significantly more than ToolACE-MT (188k). To clarify token usage, we report total input/output tokens and the corresponding cost (GPT-4o pricing: $2.5/M input, $10/M output). The main efficiency gain comes from reduced input tokens, since fewer iterations are needed to generate longer multi-turn dialogues. Models trained on ToolACE-MT data also perform better, demonstrating both higher efficiency and effectiveness.

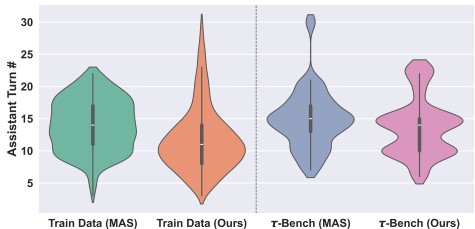 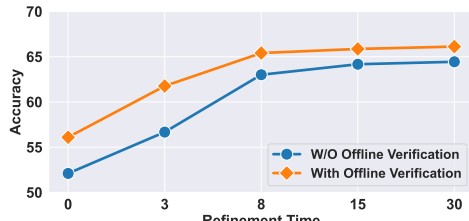

Figure 4: Statistics of assistant turn counts for MAS and our method, measured on the training data and successful inference cases in $\tau$-Bench.

Figure 5: The accuracy results of our method on BFCL-v3 when scaling Iterative Refinement times, with and without offline verification.

**Model Choices.** We further test ToolACE-MT with GPT-4o-mini (results also shown in Table 5), which results in a much lower pass rate and increased API calls, due to more frequent formatting errors and hallucinations. This reinforces that generating long, tool-intensive dialogues demands strong long-context handling, which smaller models like GPT-4o-mini and LLaMA3.1-8B-Inst (in our attempt, it failed to produce valid instances in most of time thus cannot generate sufficient usable data for training) struggle with.

Finally, even after filtering, the model trained on GPT-4o-mini generated data still show a notable performance gap compared to that trained on GPT-4o generated data ($60.13\%$ vs. $65.41\%$), highlighting that generation quality before verification remains crucial despite post-processing.

**Task Completion Efficiency.** In addition to generation efficiency, we further examine how our non-autoregressive pipeline influences task completion efficiency. Here we define task completion efficiency as how effectively the assistant completes a user task, where turn numbers per task serve as a signal. We hypothesize that this generation paradigm supports more effective overall task planning, thereby reducing the number of interaction turns required to complete a task. In contrast, MAS often involves trial-and-error behavior from the assistant model to identify correct actions. This hypothesis is first supported by statistics from the training data: as shown in the left part of Figure 4, instances generated by our method ToolACE-MT have fewer assistant turns on average than those generated by MAS. Evaluation on $\tau$-Bench (right part) further validates this advantage, where our model completes tasks successfully with an average of $13.7$ assistant turns, compared to $15.4$ turns for MAS. These findings suggest that ToolACE-MT leads to better task structuring and more efficient interaction patterns.

## 4.5 DATA EFFECTIVENESS AND GENERALIZABILITY

**Iterative Refinement Time Scaling.** In the previous subsection, we mentioned the complementary roles of Iterative Refinement and Offline Verification, both of which contribute to enhancing final data quality. To further investigate their interaction, we conduct an experiment where we vary the number of Iterative Refinement steps, specifically by applying more Reasonability Refinement operations (as Complexity Injection is not well-suited for repeated application within a single dialogue). For each refinement level, we train two models: one using data that has passed Offline Verification and one without. The performance trends are illustrated in Figure 5.

As shown, when the number of refinement steps is low, the performance gap between models trained with and without Offline Verification is large (around $5\%$), indicating that Offline Verification is crucial for filtering low-quality data in the pipeline. As refinement iterations increase, this gap narrows (dropping below $2\%$ after 15 iterations), showing that additional refinement improves data quality and reduces the need for further filtering. However, the gap never fully disappears even after 30 refinement steps, highlighting the distinct but complementary roles of the two stages. While Iterative Refinement primarily improves semantic coherence and function call accuracy, Offline Verification excels at catching issues like long-range inconsistencies or overall structural flaws that are harder to correct through refinement alone.

**Different Backbones.** To evaluate the generalizability of our generated data across different backbone models, we conduct experiments using base models of similar sizes, including Qwen2.5-7B-Inst and Qwen3-8B (no-thinking). The results, shown in Figure 6, include comparisons between raw

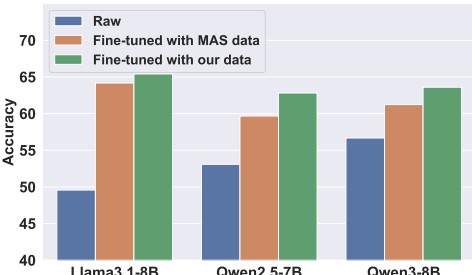 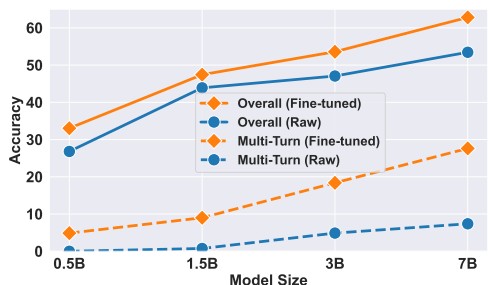

Figure 6: The accuracy results on BFCL-v3 when training based on different backbones.

Figure 7: The accuracy results on BFCL-v3 for Qwen2.5-Inst series models.

models (without training), models trained with MAS data, and models trained with ToolACE-MT. As observed, both backbones benefit from training with MAS data, and training with ToolACE-MT leads to further consistent gains.

Interestingly, although the initial (raw) performance of Qwen2.5-7B-Inst and Qwen3-8B is higher than that of Llama3.1-8B-Inst, the performance gain after fine-tuning is smaller. We attribute this to the training strategies of more recent models. Both Qwen2.5 and Qwen3 were released after Llama3.1 and are likely to have incorporated improved agentic capabilities during their training. As a result, further fine-tuning on similar task formats may yield diminishing returns, reflecting a saturation effect from repeated exposure to related domains.

**Model Size Scaling.** Scaling laws suggest a strong correlation between model size and performance. To explore the scalability of function calling capabilities after training on our generated data, we evaluate the Qwen-2.5-xB-Inst series across a range of model sizes (0.5B, 1.5B, 3B, and 7B). Both the raw and fine-tuned versions (trained on our generated 8000 instances) are assessed on BFCL-v3, with results (Multi-Turn and overall) shown in Figure 7. As expected, larger models consistently outperform smaller ones. The smaller raw models (0.5B and 1.5B) exhibit little to no multi-turn capabilities, but fine-tuning with our dataset can enhance the corresponding performance. Notably, the improvements are more pronounced in the 3B and 7B models, suggesting that multi-turn function calling remains a relatively advanced ability that small models struggle to acquire. Overall, the fine-tuned models demonstrate a clear scaling trend, reinforcing the effectiveness of our data in equipping larger LLMs with complex function calling skills.

## 5 CONCLUSION

This paper introduces ToolACE-MT, a non-autoregressive framework for generating multi-turn function-calling dialogues. Inspired by non-autoregressive generation, ToolACE-MT combines iterative refinement and offline verification to ensure semantic coherence, contextual consistency, and tool executability. It achieves substantial improvements in multi-turn function-calling accuracy, outperforming strong baselines while being efficient in both data generation and task completion. Further analysis demonstrates the complementary effects of refinement and verification, as well as the generalizability of ToolACE-MT across various model sizes and backbones.

## REPRODUCIBILITY STATEMENT

We have taken several measures to ensure the reproducibility of our work. The proposed methods and experimental settings (including data amount, models used for data generation, configuration during data generation, trained models, evaluation details, and hyper-parameters for training) are described in detail in the main text (Sections 3 and 4.1), with additional implementation details provided in the appendix, including hyper-parameter tuning strategies, prompts used for data generation, and training protocols. We also provide a representative data example in the appendix to illustrate the structure and format of the data. Together, these efforts are intended to allow other researchers to replicate our experimental setup and verify our findings with minimal ambiguity.

ACKNOWLEDGMENTS

The research described in this paper is partially supported by The National Natural Science Foundation of China (Grant No. 62502118, 62502310).

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

## A   MORE TRAINING DETAILS

Table 6 displays the selections of hyper-parameter, where all are selected based on performance of development set. All the models are trained based on LLaMA-Factory[2].

Table 6: Hyper-parameters in experiments.

| Hyper-parameters | Selection Ranges | Final Selction |
|---|---|---|
| **Learning rate** | [5e-5, 1e-4, 2e-4] | 1e-4 |
| **Batch size** | [32, 64] | 64 |
| **Warmup ratio** | [0.1] | 0.1 |
| **Lora rank** | [8, 16] | 16 |
| **Lora alpha** | [16, 32] | 32 |

## B   USE OF LLMS

We used large language models (LLMs) solely for writing assistance, specifically to polish the language of the manuscript. The models were employed to improve readability, grammar, and style, without altering the technical content, analysis, or results. All ideas, experimental designs, and conclusions were conceived and developed by the authors.

## C   FULL RESULTS ON ACEBENCH

Full results on ACEBench are shown in Table 7, where additional results for single-turn categories are added ("S-Turn" is short for Single-Turn, "M-Turn" is short for Multi-Turn). The results reflect that ToolACE-MT still outperforms baselines in those single-turn categories, with a higer overall result.

Table 7: Accuracy (%) comparison on ACEBench (En) full set.

| Models | Atom | S-Turn | M-Turn | Similar | Preference | Special | Agent (EA) | Agent (PA) | Overall |
|---|---|---|---|---|---|---|---|---|---|
| **GPT-4o-2024-11-20** | 90.0 | 78.0 | 68.0 | 80.0 | 78.0 | 92.7 | 56.0 | 77.8 | 81.1 |
| **Llama3.1-70B-Inst** | 83.7 | 71.5 | 61.0 | 74.0 | 66.0 | 29.3 | 41.0 | 62.5 | 57.9 |
| **Llama3.1-8B-Inst** | 52.7 | 45.0 | 24.0 | 54.0 | 50.0 | 9.3 | 6.7 | 18.3 | 30.4 |
| **Multi-Agent Simulation** | 81.3 | 63.5 | 48.0 | 70.0 | 64.0 | 5.3 | 6.7 | 15.0 | 43.8 |
| **ToolACE-MT** | 83.0 | 64.0 | 51.0 | 68.0 | 68.0 | 8.7 | 8.4 | 34.0 | 45.2 |
| - Offline Verification | 77.1 | 61.0 | 44.0 | 60.0 | 64.0 | 8.7 | 1.7 | 28.5 | 41.8 |
| - Iterative Refinement | 61.7 | 56.0 | 34.0 | 56.0 | 50.0 | 5.3 | 1.7 | 22.8 | 34.5 |

## D   ADDITIONAL EXPERIMENTS

To further highlight the value of our generated data, we conducted additional experiments on three representative open-source datasets (ToolACE-10k[3], Toucan-1.5M[4], ToolDial-9k[5]), where we train several models using the same training setting and the corresponding results on the three benchmarks are summarized in Table 8. As shown, except for the $\tau$-Airline dataset (for which we previously noted reliability issues), our ToolACE-MT consistently achieves superior performance across all multi-turn evaluations. Although it lags slightly behind ToolACE (which primarily contains single-turn samples) in the single-turn setting, ToolACE-MT still delivers substantial improvements over

---

[2]https://github.com/hiyouga/LLaMA-Factory
[3]https://huggingface.co/datasets/Team-ACE/ToolACE
[4]https://huggingface.co/datasets/Agent-Ark/Toucan-1.5M
[5]https://github.com/holi-lab/ToolDial/tree/main

the base Llama3.1-8B model. Overall, the results indicate that the data generated by ToolACE-MT matches or surpasses that produced by existing methods, with particularly strong gains on multi-turn scenarios, underscoring the effectiveness and superiority of our ToolACE-MT approach.

Table 8: Accuracy comparison (%) on the three representative benchmarks.

| Models | BFCL | | ACEBench | | $\tau$-Bench | |
|---|---|---|---|---|---|---|
| | *Single-Turn* | *Multi-Turn* | *Multi-Turn* | *Agent-PA* | *$\tau$-Retail* | *$\tau$-Airline* |
| **Llama3.1-8B-Inst** | 72.7 | 9.3 | 24.0 | 18.3 | 6.1 | 26.0* |
| **+ ToolACE-10k** | 80.0 | 8.6 | 51.0 | 14.0 | 13.9 | 22.0 |
| **+ Toucan-1.5M (sampled 8k)** | 75.2 | 15.0 | 41.0 | 22.6 | 18.3 | 18.0 |
| **+ ToolDial-9k** | 70.1 | 12.3 | 30.0 | 11.4 | 10.4 | 20.0 |
| **+ MAS (this work)** | 79.2 | 31.4 | 48.0 | 15.0 | 21.7 | 10.0 |
| **+ ToolACE-MT (this work)** | 78.2 | 40.3 | 51.0 | 34.0 | 25.2 | 16.0 |

## E  FAILURE CASES ANALYSIS

This section analyzes the most frequent failure modes encountered during Offline Verification. Table 9 summarizes the top three failure types for both rule-based and model-based verification. As shown, most errors stem from parameter-related issues, including incorrect parameter values, mismatches between parameters and function signatures, and improper or incomplete parameter formatting. These findings highlight that parameter handling remains the primary bottleneck in achieving fully reliable tool-calling data generation.

Table 9: Top-3 failure problems for Offline Verification.

| Rule-based Problems | Ratio | Model-based Problems | Ratio |
|---|---|---|---|
| **Parameter Hallucination** | 51% | **Parameter Extraction** | 23% |
| **No Tool Calling** | 18% | **Tool Chosen/Ordering** | 21% |
| **Conversation Role Misordering** | 11% | **Parameter Format Compliance** | 16% |

## F  DATA EXAMPLE

Below we show a data example for reference.

**[system]** You are an expert in composing functions. You are given a question and a set of possible functions. Based on the question, you will need to make one or more function/ tool calls to achieve the purpose. If none of the functions can be used, point it out. If the given question lacks the parameters required by the function, also point it out. You should only return the function calls in your response. If you decide to invoke any of the function(s), you MUST put it in the format of [func_name1(params_name1=params_value1, params_name2=params_value2...), func_name2(params), ...] You SHOULD NOT include any other text in the response. Here is a list of functions in JSON format that you can invoke:
{"name": "get_curr_date", "description": "Retrieves the current date in YYYY-MM-DD format.", "parameters": ...}
{"name": "get_curr_weekday", "description": "Retrieves the current day of the week.", "parameters": ...}
{"name": "book_flight", "description": "Books a flight between two cities on a given date.", "parameters": ...}
...

---

**[user]** I'm planning a trip and need to fly to Shanghai. Could you help me book a flight for next Tuesday?

---

**[assistant]** Could you provide your departure city?

---

**[user]** Singapore.

---

**[assistant]** [get_curr_date(), get_curr_weekday()]

---

**[tool]** [{"current_date": "2023-10-05"}, {"current_ weekday": "Thursday"}]

---

**[assistant]** [book_flight(departure_city='Singapore', arrival_city='Shanghai', departure_date='2023-10-10')]

---

**[tool]** [{"confirmation": "Flight successfully booked from Singapore to Shanghai on 2023-10-10."}]

---

**[assistant]** Great news! Your flight from Singapore to Shanghai for next Tuesday, October 10th, has been successfully booked. Feel free to let me know if there's anything else you need assistance with!

---

**[...]** ... (More Turns)

---

The data highlights several essential abilities that an agent must possess. For example:

1) Asking clarification questions when user input is ambiguous or incomplete.

2) Multi-step function calling by leveraging implicit cues from historical context. Here the assistant needs to book a flight, but the book_flight tool requires a specific date in YYYY-MM-DD format. As the date is not provided explicitly, the agent should infer it using supporting tools including get_curr_date and get_curr_weekday.

## G  PROMPTS FOR DATA GENERATION

Figure 8 to Figure 15 display the prompts we use for our ToolACE-MT data generation.

You are a task generation expert. Your responsibility is to generate a multi-step, tool-usage-related task description in English, based on the given inputs following the requirements.

You will be provided with:

- Several examples for your reference;
- A list of available tool candidates;
- One or more completed task descriptions (may also be empty);
- A target number of steps N, indicating that the new task should contain N sequential tool calling steps.

## Task Structure Requirements

1. Write a concise paragraph in English that describes a complete objective consisting of multiple logically related subtasks.
2. The task should contain N steps that can be executed sequentially, with each step triggering one or more tool callings.
3. Parallel tool callings (e.g., processing multiple unrelated callings independently at the same time) are counted as a single step.
4. The steps should exhibit contextual dependency or natural progression, forming a coherent task flow.
5. Each step can be described at an abstract level (no need for detailed parameters), but the executable intent must be clear.

## Continuation Requirements

- If the "Completed Task" input is not empty, your newly generated task should serve as a natural continuation of those tasks, such as further processing, analysis, or expansion within the same context or based on the existing results.
- If the "Completed Task" input is empty, you are free to invent a reasonable new task flow.

## Language Requirements

- The output should be an English task description.
- The description should be concise and fit the context of multi-turn tool usage.

## Given Inputs
### Task Examples
{examples}

### Available Tool Candidates
{candidate_tools}

### Completed Task
{completed_task}

### Target Step Number
{step_number}

## Output Format
<Task_Start>... (English task description)<Task_End>

Figure 8: The prompt for task initialization.

You are a multi-turn tool-calling dialogue completion expert. Your responsibility is to simulate the complete trajectory for given task description, based on the given inputs following the requirements.

You will be provided with:

- One example trajectory for your reference;
- A list of available tool candidates;
- Current task description;
- History trajectory that about the previous task completion.

## Completion Requirements

1. The trajectory should start with the user role raising a request, followed by the assistant role completing the task interacting with the tool role. The final turn should be the assistant role, summarizing all results to the user role.

2. The user role should avoid direct descriptions of operation steps. Instead, the requests should be embedded in context with appropriate discourse markers, interjections, and connecting language to better resemble real human interaction.

3. The user input should provide complete parameter information required for tool invocation.

4. The format for the assistant role to call the tools is: $[func\_name1(params\_name1 = params\_value1, params\_name2 = params\_value2...), func\_name2(params)]$, followed by a tool turn returning results.

5. Tool return results must be in dictionary format, based on the calling parameters in the preceding assistant turn and the tool's functionality introduced in tool description.

## Language Requirements

- The output should be in English.
- The whole trajectory should be reasonable and fit the context of multi-turn tool usage.

## Given Inputs
### Example Trajectory
{example}

### Available Tool Candidates
{candidate_tools}

### Current Task
{current_task}

### History Trajectory
{history_trajectory}

## Output Format

```
[
    {"role": "user", "content": "..."},
    {"role": "assistant", "content": "..."},
    {"role": "tool", "content": "..."},
    ...
]
```

Figure 9: The prompt for trajectory initialization.

You are a data transformation expert. Your responsibility is to modify and extend one specific user turn in a given conversation, following the requirements.

You will be provided with:

- One example for your reference;
- A list of available tool candidates;
- A conversation need to be modified;
- The specific user turn to be modified and extended.

## Specific requirements

1. First, modify the user's content in this turn to make it a vague question or omit necessary information, so that the assistant cannot determine which tool to use or lacks the required parameters needed to invoke the tool (avoid using 'this', but 'a' or 'some').

2. Then, extend the conversation by adding an assistant turn that asks questions (the assistant cannot assume prior knowledge of the user's intent; the question should naturally match the context) to gather sufficient information for invoking the tool.

3. After that, extend with a user turn that provides a complete and accurate answer with the required parameters.

4. Ensure that the modified and extended conversation remains smooth, natural, and reasonable.

## Given Inputs
### Example Modification
{example}

### Available Tool Candidates
{candidate_tools}

### Given Conversation
{conversation}

### Target User Turn
{user_turn}

## Output Format

```
[
    {"role": "user", "content": "..."},
    {"role": "assistant", "content": "..."},
    {"role": "user", "content": "..."}
]
```

Figure 10: The prompt for adding clarification turns in complexity injection.

You are a data transformation expert. Your responsibility is to extend one specific user turn in a given conversation, following the requirements.

You will be provided with:

- One example for your reference;
- A list of available tool candidates;
- A conversation need to be modified;
- The specific user turn to be extended.
- The specific candidate tool to be removed.

## Specific requirements

1. Keep the user turn entirely unchanged, but adding two additional turns.
2. The first added turn should be an assistant turn, expressing that the current candidate tools cannot meet the user's needs.
3. The second added turn should be a user turn, directly providing the description of the removed tool for the assistant to call.
4. Ensure that the extended conversation remains smooth, natural, and reasonable.

## Given Inputs
### Example Extension
{example}

### Available Tool Candidates
{candidate_tools}

### Given Conversation
{conversation}

### Target User Turn
{user_turn}

### The Tool to be Removed
{removed_tool}

## Output Format

```
[
    {"role": "user", "content": "..."},
    {"role": "assistant", "content": "..."},
    {"role": "user", "content": "..."}
]
```

Figure 11: The prompt for tool awareness in complexity injection.

You are a data transformation expert. Your responsibility is to extend one specific assistant turn in a given conversation, following the requirements.

You will be provided with:

- One example for your reference;
- A list of available tool candidates;
- A conversation need to be modified;
- The specific assistant turn to be extended.

## Specific requirements

1. Modify the tool calling part of the assistant turn, injecting one error parameter value.
2. Add a tool turn returning error messages and showing possible solutions.
3. Then add another assistant turn that corrects the tool calling statement.
4. Ensure that the modified and extended conversation remains smooth, natural, and reasonable.

## Given Inputs
### Example Modification
{example}

### Available Tool Candidates
{candidate_tools}

### Given Conversation
{conversation}

### Target Assistant Turn
{assistant_turn}

## Output Format

```
[
    {"role": "assistant", "content": "..."},
    {"role": "tool", "content": "..."},
    {"role": "assistant", "content": "..."}
]
```

Figure 12: The prompt for error simulation in complexity injection.

You are a data transformation expert. Your responsibility is to modify and extend one specific user turn in a given conversation, following the requirements.

You will be provided with:

- One example for your reference;
- A list of available tool candidates;
- A conversation need to be modified;
- The specific user turn to be modified and extended.

## Specific requirements

1. Add two turns before the specific user turn.

2. The first added turn should be a user turn. Its content may be casual chit-chat or a request that does not require function calling (e.g., asking for recommendations, translation, or open-ended writing). The topic should be related to the original user turn.

3. The second added turn should be an assistant response directly addressing the first added user turn.

4. Keep the content of the original (specified) user turn unchanged, and append it as the next turn.

5. Ensure that the modified and extended conversation remains smooth, natural, and reasonable.

## Given Inputs
### Example Modification
{example}

### Available Tool Candidates
{candidate_tools}

### Given Conversation
{conversation}

### Target User Turn
{user_turn}

## Output Format

```
[
    {"role": "user", "content": "..."},
    {"role": "assistant", "content": "..."},
    {"role": "user", "content": "..."}
]
```

Figure 13: The prompt for non-function-calling in complexity injection.

You are a data completion expert. Given a conversation between a user and an assistant, where the assistant can perform tool calling to complete the user's task, your responsibility is to fill in the missing content following the requirements.

You will be provided with:

- A list of available tool candidates;
- A partially completed conversation, with some content missing and replaced by placeholders such as `"xxx"`, `"yyy"`, etc.

## Completion Requirements

1. You should try your best to recover the missing content, by replacing the placeholders with actual content.

2. If the recovered content is in a user turn, the content should avoid direct descriptions of operation steps. Instead, the requests should be embedded in context with appropriate discourse markers, interjections, and connecting language to better resemble real human interaction.

3. If the recovered content is in an assistant turn and need calling tools, the format for the assistant role to call the tools is: $[func\_name1(params\_name1 = params\_value1, params\_name2 = params\_value2...), func\_name2(params)]$.

4. If the recovered content is in a tool turn, you should simulate a reasonable tool output that coherent with its adjacent turns' actions.

5. Ensure that the recovered whole conversation is smooth, natural, and reasonable.

## Given Inputs
### Available Tool Candidates
{candidate_tools}

### Given Conversation
{conversation}

## Output Format
```
{
    "xxx": "...",
    "yyy": "...",
    ...
}
```

Figure 14: The prompt for mask-and-fill in reasonability refinement.

You are a data quality evaluation expert. Given a conversation history and two possible continued trajectories, your responsibility is to determine which continued trajectory is of higher quality.

You will be provided with:

- A list of available tool candidates;
- A conversation history;
- Two continued trajectories.

## Evaluation Criteria

1. Coherence: Choose the trajectory that exhibits smooth and natural progression.
2. Correctness: Tool calling statements must be strictly correct, consistent with the dialogue history, and must not assume any values that have not previously appeared.
3. Consistency: Pay close attention to aspects such as user-assistant consistency, the plausibility of parallel function calls, tool output formatting, and overall structure.
4. Deep thinking: Before providing your final judgment, first present your reasoning process.

## Given Inputs
### Available Tool Candidates
{candidate_tools}

### Given Conversation History
{conversation}

### Continued Trajectory A
{trajectory_a}

### Continued Trajectory B
{trajectory_b}

## Output Format

```
{
    "think": "...",
    "judgement": "A/B",
}
```

Figure 15: The prompt for judger in reasonability refinement.

