# OpenReview forum: "ToolACE-MT: Non-Autoregressive Generation for Agentic Multi-Turn Interaction"
_ICLR.cc/2026/Conference — ICLR 2026 Poster_

### Official Review · Reviewer_JVVr · 2025-10-27

**Soundness:** 2
**Presentation:** 3
**Contribution:** 3
**Rating:** 6
**Confidence:** 5

**Summary:**

This paper proposes a non-autoregressive iterative generative framework (NAIG) for constructing high-quality multi-agent dialogues. NAIG consists of three main phases: coarse-grained dialogue structure initialization, iterative refinement, and offline verification. NAIG first globally initializes a fully structured multi-agent dialogue. Iterative refinement uses masking and padding to continuously refine dialogue details. The offline verification phase verifies the correctness and consistency of the dialogue. Experiments show that, using the same base model, training on data generated by NAIG achieves better accuracy and efficiency than traditional multi-agent simulations.

**Strengths:**

This work makes the following contributions to synthesizing complex agent task data:
(1) It proposes a framework for generating complex agent task data, focusing on refining conversation details while maintaining the overall task through masking and padding.
(2) It splits initialization and refinement into two phases, allowing for the definition and selection of task complexity to be completed at a low cost in the initialization phase, while controlling complexity and budget by the number of iterations in the iterative refinement phase.
(3) The NAIG framework offers a straightforward approach that addresses the following motivations: (a) When using autoregressive methods, trajectories often exhibit undesirable shifts (especially in long-range scenarios). Employing the mask-fill method allows LLMs to perceive the full context of trajectories, mitigating inconsistencies arising from modifications to some extent. (b) During initial trajectory generation, NAIG enables explicit control over length and complexity, thereby facilitating more controllable data design.

**Weaknesses:**

(1) The article mentions (Table 4) that NAIG is less expensive than traditional MAS. I understand this refers to the number of API calls. Although the number of API calls is less than MAS, since operations such as iterative optimization may consume a lot of tokens, is the total token cost still lower than MAS?
(2) In Section 4.3, the authors mention that there is a significant performance gap between GPT-4o-mini and GPT-4o. This highlights that the quality of initial generation is still crucial. I don't understand why only the quality of initial generation is mentioned here.
(3) I would like to know the specific steps of the MAS experiment in the article. How is it different from NIAG, which only uses the initialization phase and offline verification phase?

**Questions:**

(1) Is there comparative data on the total token consumption for NAIG and MAS?
(2) I would like to know the specific steps of the MAS experiment in the article. How is it different from NIAG, which only uses the initialization phase and offline verification phase?

---

> ### Author Response · Authors · 2025-11-19
> **Author Response**
>
> **W1&Q1: The article mentions (Table 4) that NAIG is less expensive than traditional MAS. I understand this refers to the number of API calls. Although the number of API calls is less than MAS, since operations such as iterative optimization may consume a lot of tokens, is the total token cost still lower than MAS?**
>
> **A:** Thank you for raising this concern. To address it, we have calculated the token consumption (separating input and output tokens) and the corresponding cost for both methods when generating the final 8K valid samples. Below are the statistics:
>
> | Method | Input Tokens (#) | Output Tokens (#) | Total Cost (GPT-4o Pricing, USD) |
> |:--|:--:|:--:|:--:|
> | MAS |652,145k | 10,712k | 1737 |
> | NAIG | 478,115k | 18,480k | 1380 |
>
> As shown, NAIG requires more output tokens, but its total cost is lower. This is calculated using OpenAI's official pricing, where input tokens cost `$`2.5 per million and output tokens cost `$`10 per million. The primary advantage of NAIG lies in its higher pass rate for offline verification (72.3% vs. 61.1%, as shown in Table 4), meaning it needs fewer raw samples to achieve the same number of valid final samples. Additionally, NAIG demonstrates similar or even better input token consumption per dialogue, as it requires fewer iterations to produce longer, multi-turn dialogues.
>
>
> **W2: In Section 4.3, the authors mention that there is a significant performance gap between GPT-4o-mini and GPT-4o. This highlights that the quality of initial generation is still crucial. I don't understand why only the quality of initial generation is mentioned here.**
>
> **A:** We apologize for the confusion. Here, "initial generation quality" refers to the entire data quality before offline verification, not only the initialization stage. In Section 4.3, our point is that even after applying quality-filtering process, a clear performance gap remains between data generated by GPT-4o-mini and GPT-4o. This indicates that all earlier generation stages (i.e., every stage prior to offline verification) still play a critical role. Moreover, the final model performance depends not only on correctness but also on properties such as diversity and complexity, aspects that offline verification alone cannot fully evaluate or enforce. These characteristics are strongly influenced by the capability of the model used for generation, which explains why performance differences persist even after filtering.
>
>
> **W3&Q2: I would like to know the specific steps of the MAS experiment in the article. How is it different from NAIG, which only uses the initialization phase and offline verification phase?**
>
> **A:** MAS uses the same tool candidate pool and the same offline verification phase as NAIG. The key difference lies in how the tool-calling trajectories are produced. For NAIG, once the task is initialized, an entire trajectory, e.g., (U, A, T, A, T) can be generated in a single API call. In contrast, MAS generates trajectories in a back-and-forth multi-agent simulation involving three agents -- User, Assistant, and Tool. Each turn in the trajectory requires a separate model interaction. For example, for a trajectory such as (U, A, T, A, T):
>
> - The User agent produces a query based on the current task plan.
> - The Assistant agent responds, including making tool-calling decisions.
> - The Tool agent (or a real tool) returns the tool output.
> - The process continues turn by turn until the full trajectory is formed.
>
> Thus, while NAIG can generate the sequence in one shot, MAS requires five separate calls for this example, illustrating the core procedural difference between the two approaches. However, one-shot generation may lead to low-quality trajectories. So we design iterative refinement process for better quality acquisition, which is motivated by mask-and-fill operations in non-autoregressive generation.

---

> > ### Author Response · Authors · 2025-11-26
> >
> > Dear Reviewer,
> >
> > We kindly invite you to review our response and, if you feel your concerns have been adequately addressed, to consider updating your score. If you have any remaining questions, we would be very happy to provide further clarification. Thank you sincerely for your time and thoughtful evaluation!

---

> ### Comment · Reviewer_JVVr · 2025-11-26
>
> Thanks for the authors' response. I will maintain my current positive rating.

---

> > ### Author Response · Authors · 2025-11-27
> >
> > Thank you! We are welcome to have more disscusion if you have remaining concerns.

---

### Official Review · Reviewer_9xQr · 2025-10-31

**Soundness:** 3
**Presentation:** 3
**Contribution:** 3
**Rating:** 6
**Confidence:** 4

**Summary:**

This paper proposes Non-Autoregressive Iterative Generation for Agentic Multi-Turn Interaction. The authors address the need for high-quality multi-turn, multi-step dialogues involving function calls and user-agent interaction in tool-augmented LLM settings. They argue that existing multi-agent simulation approaches rely on costly autoregressive generation across multiple agents. To improve efficiency and control, they introduce a three-stage data construction framework: coarse-grained initialization, iterative refinement, and offline verification. The initialization produces a structurally complete but semantically rough dialogue skeleton; the refinement injects complexity and uses mask-and-fill operations to improve the trajectories; the verification filters for correctness and coherence via rule-based and model-based checks. Experiments on multiple benchmarks show that models fine-tuned on data generated by their framework outperform MAS-generated data in downstream function-calling tasks while reducing API-call cost.

**Strengths:**

- The paper addresses an important problem: generating realistic, tool-augmented agentic dialogues for training LLMs with function-calling and multi-turn behaviour.
- The framework is clearly presented. The three-stage design offers a practical pipeline that balances structure, refinement and quality control.
- The experimental evaluation is reasonably broad. Multiple benchmarks, different model backbones, cost vs quality trade-off are all considered.
- The authors attempt to quantify cost savings (via API-call count) and show downstream performance improvements, which strengthens the applied relevance of the work.

**Weaknesses:**

1. Lack of human evaluation and direct quantitative assessment of the generated data itself. The paper focuses on downstream model performance improvements, but fails to report metrics such as diversity (Distinct-n), structural repetition (how many dialogue skeleton templates were reused), naturalness/human preference, or other data-intrinsic quality measures.
2. Cost evaluation metric is limited. While the paper reports that API calls drop from ≈ 275k (MAS) to ≈ 188k (their method), there is no breakdown of token consumption, compute hours, or monetary cost. Thus the claim of “cost reduction” lacks transparency and might not reflect real-world resource savings.
3. Failure case analysis could be provided. The paper does not include a detailed look at failed/generated invalid samples: which trajectories failed verification, what error modes occurred (parameter mismatch, tool-call misordering, semantic incoherence), or how many were discarded. Without this, it is hard to judge the robustness or failure modes of the proposed pipeline.
4. The initialization stage lacks transparency and poses risks of structural homogenization or unrealistic data distribution. The “coarse-grained initialization” phase is described at a high level but lacks detail on how skeletons are generated, how varied they are, whether templates are reused, or whether sub-tasks are meaningfully correlated as in real distributions. This raises concerns that many samples might share the same skeleton, reducing diversity, or that sub-tasks may be arbitrarily concatenated in ways rare in real user-agent dialogues.
5. The novelty is somewhat incremental. The combination of non-autoregressive generation + iterative mask-and-fill + simulation for data generation is practically useful, but from a model-research perspective, it is more of a pipeline engineering effort than a conceptual leap in generation modelling.

**Questions:**

1. Can you provide quantitative metrics on the generated dataset itself: e.g., number of unique skeleton templates used, duplication rate of structures, Distinct-n diversity scores, or human preference judgments?
2. Could you report token counts, compute hours and/or monetary cost estimates for the data generation process (initialization + refinement + verification) to strengthen the cost-reduction claim?
3. Do you have statistics on how many samples were discarded during offline verification, and what the major categories of failure were? Could you share sample failure cases?
4. How do you ensure that sub-task sequences in each dialogue reflect realistic correlations rather than random concatenations? Did you compare your generated task‐structure distribution with that of real agent dialogues?
5. Are there template or skeleton reuse effects in the initialization phase? How many skeleton variants were used, and how did you guard against structure homogenization?

---

> ### Author Response · Authors · 2025-11-19
> **Response Part 1**
>
> **W1&Q1: Lack of human evaluation/quantitative assessment of the generated data itself. Can the author provide information like number of unique skeleton templates used, duplication rate of structures, Distinct-n diversity scores, or human preference judgments, etc.?**
>
> **A:** Thank you for raising this concern. We have conducted additional data statistics and quality analysis, and the results are summarized below:
>
> | Method | Avg tool # per dialogue | Avg turn # per dialogue | Avg user turn # per dialogue |Avg call turn # per dialogue| Avg call # per user turn | Hand-craft example # |
> |:--|:--:|:--:|:--:|:--:|:--:|:--:|
> | MAS | 8.8 | 28.0 | 5.8 | 8.7 | 2.3 | 10 |
> | NAIG | 8.4 | 23.7 | 3.4 | 8.5 | 3.7 | 10 |
>
> The "Hand-Crafted Example #" column refers to the original examples manually written for initialization and complexity injection (see Figures 8-13 in the appendix for details on these prompts). During generation, more samples are introduced as reference examples if they pass offline verification. These examples can be considered analogous to the "unique skeleton templates" mentioned, though we do not explicitly use skeleton templates but rather reference examples.
>
> Additionally, following the approach of ToolFlow [1], we performed a data quality analysis using four metrics: semantic similarity (SS), ratio of entailment (EnR), Shannon entropy (H), and Distinct-3 (Dist-3). The former two are for coherence, and the latter two are for diversity. The results are as follows:
>
> | Method | SS | EnR | H | Dist-3 |
> |:--|:--:|:--:|:--:|:--:|
> | MAS | 65.23 | 43.60 | 7.92 | 0.319 |
> | NAIG | 68.34 | 50.71 | 9.28 | 0.357 |
>
> These results indicate that NAIG-generated data exhibits higher coherence and diversity compared to MAS. While time constraints prevented us from collecting human annotations, prior ToolFlow's original analysis has demonstrated strong agreement between these metrics and human judgments. Therefore, we believe these metrics are sufficient to demonstrate the quality of the generated data.
>
> [1] (Wang et al. 2025) ToolFlow: Boosting LLM Tool-Calling Through Natural and Coherent Dialogue Synthesis
>
>
> **W2&Q2: Cost evaluation metric is limited to API call numbers. Could you report token counts, compute hours and/or monetary cost estimates for the data generation process (initialization + refinement + verification) to strengthen the cost-reduction claim?**
>
> **A:** Thank you for raising this concern. To address it, we have calculated the token consumption (separating input and output tokens) and the corresponding cost for both methods when generating the final 8K valid samples. Below are the statistics:
>
> | Method | Input Tokens (#) | Output Tokens (#) | Total Cost (GPT-4o Pricing, USD) |
> |:--|:--:|:--:|:--:|
> | MAS |652,145k | 10,712k | 1737 |
> | NAIG | 478,115k | 18,480k | 1380 |
>
> As shown, NAIG requires more output tokens, but its total cost is lower. This is calculated using OpenAI's official pricing, where input tokens cost `$`2.5 per million and output tokens cost `$`10 per million. The primary advantage of NAIG lies in its higher pass rate for offline verification (72.3% vs. 61.1%, as shown in Table 4), meaning it needs fewer raw samples to achieve the same number of valid final samples. Additionally, NAIG demonstrates similar or even better input token consumption per dialogue, as it requires fewer iterations to produce longer, multi-turn dialogues.

---

> ### Author Response · Authors · 2025-11-19
> **Response Part 2**
>
> **W3&Q3: Failure case analysis could be provided. The paper does not include a detailed look at failed/generated invalid samples. Do you have statistics on how many samples were discarded during offline verification, and what the major categories of failure were? Could you share sample failure cases?**
>
> **A:** Thank you for raising this important point. While we did not delve into the specifics of offline verification in the original paper, we want to clarify that this aspect was not the focus of our contribution. Many prior works have already explored diverse verification processes for data filtering. To ensure fairness, we applied the same offline verification procedure to both MAS and NAIG in our experiments.
>
> Nevertheless, we agree that failure analysis would enhance the persuasiveness of our work. The statistics on discarded samples during offline verification were included in Table 4 of our paper, where the pass rate for NAIG is 72.3%, meaning 27.7% of samples were discarded. Additionally, we categorized the most common failure modes for both rule-based and model-based verification (top 3, respectively). Below are the statistics:
>
> | Rule-based Problems | Ratio | Model-based Problems | Ratio |
> |:--|:--:|:--|:--:|
> | Parameter Hallucination | 51% | Parameter Extraction | 23% |
> | No Tool Calling | 18% | Tool Chosen/Ordering | 21% |
> | Conversation Role Misordering | 11% | Parameter Format Compliance | 16% |
>
> As shown, the main issues involve parameters -- either incorrect values or improper formats. Below is an example failure case related to parameter hallucination, where the assistant generates a parameter value ("summary_report.txt") that does not come from the user but is instead generated by the model itself:
>
> ```
> [
>     {
>         "role": "user",
>         "content": "I need to analyze all the text files in this directory and its subdirectories. Could you help me search for all .txt files first, then count the lines in each one? After that, I'd like to create a summary report with the file names and their line counts, and finally sort that report alphabetically by filename. Let me start by finding all the text files around here."
>     },
>     {
>         "role": "assistant",
>         "content": "[find(name=\".txt\")]"
>     },
>     {
>         "role": "tool",
>         "content": "{'matches': ['file1.txt', 'file2.txt', 'file3.txt']}"
>     },
>     {
>         "role": "assistant",
>         "content": "[wc(file_name=\"file1.txt\", mode=\"l\"), wc(file_name=\"file2.txt\", mode=\"l\"), wc(file_name=\"file3.txt\", mode=\"l\")]"
>     },
>     {
>         "role": "tool",
>         "content": "[{'count': 10, 'type': 'lines'}, {'count': 15, 'type': 'lines'}, {'count': 8, 'type': 'lines'}]"
>     },
>     {
>         "role": "assistant",
>         "content": "[echo(content=\"File Line Count Summary\\n\\nfile1.txt: 10 lines\\nfile2.txt: 15 lines\\nfile3.txt: 8 lines\", file_name=\"summary_report.txt\")]"
>     },
>     ...
> ]
> ```
>
> We believe that these insights, along with the detailed statistics on failures, provide a clearer understanding of the challenges faced during data generation and the types of errors that arise.

---

> ### Author Response · Authors · 2025-11-19
> **Response Part 3**
>
> **W4: The initialization stage lacks transparency and poses risks of structural homogenization or unrealistic data distribution. The “coarse-grained initialization” phase is described at a high level but lacks detail on how skeletons are generated, how varied they are, whether templates are reused, or whether sub-tasks are meaningfully correlated as in real distributions. This raises concerns that many samples might share the same skeleton, reducing diversity, or that sub-tasks may be arbitrarily concatenated in ways rare in real user-agent dialogues.**
>
> **A:** We appreciate the reviewer’s concern regarding the transparency and realism of the initialization stage. Below we provide additional clarification and justification.
>
> The initialization stage is designed to construct a coarse-grained sample where tasks together with corresponding naive tool-calling trajectories are generated. Rather than relying on fixed templates, the process generates each sub-task and its trajectory sequentially, conditioning on all previously generated steps. Such construction prevents rigid skeleton reuse and ensures that each trajectory evolves organically from prior context instead of being stitched together arbitrarily.
>
> To avoid structural homogenization, every initialization run is exposed to different combinations of 1) reference examples, 2) available tool candidates, and 3) target step-length constraints. This results in diverse high-level task structures. The exact prompt configurations for these components are provided in Figures 8–9 of the appendix. Importantly, we do not use fixed skeleton templates; instead, the model receives varied reference examples chosen from high-quality hand-written and verified samples. This provides structural cues without enforcing rigid reuse.
>
> Realism is encouraged in two ways: 1) High-quality hand-crafted examples used during initialization demonstrate natural sub-task correlations and realistic decomposition patterns observed in real tool-use dialogues. 2) In the complexity injection phase, we further revise the initial trajectories by introducing multiple categories of realistic complexities (e.g., incomplete user intents, error correction, etc.). This step intentionally breaks overly linear structures and injects behavioral phenomena characteristic of real-world interactions.
>
> The above together mitigate the risk of low diversity or unrealistic tasks and produce trajectories with diverse yet coherent structures.
>
> Finally, our evaluations on $\tau$-Bench and ACEBench-Agent show substantial improvements. Since these benchmarks are designed to reflect realistic tool-use patterns, the observed gains indicate that our synthesized data does not deviate from practical user–agent dialogue distributions.
>
>
> **W5: The novelty is somewhat incremental. The combination of non-autoregressive generation + iterative mask-and-fill + simulation for data generation is practically useful, but from a model-research perspective, it is more of a pipeline engineering effort than a conceptual leap in generation modelling.**
>
> **A:** First of all, we want to argue that the combination of non-autoregressive generation and multi-turn tool-calling data generation is already a **paradigm innovation**, given that most existing related work still relies on Back-and-Forth generation (i.e., multi-agent simulation in this work) [1,2,3,4]. In contrast, NAIG generates the whole trajectory in a non-autoregressive manner and then refines it, which is fundamentally different from the sequential generation assumption in prior approaches.
>
> On the other hand, for data-generation papers, most novelty typically lies in designing more effective mechanisms within a relatively standard pipeline. For example, ToolFlow [1] and MAGNET [2] introduce graph-related operations for producing higher-quality data, but remain within the same back-and-forth paradigm. In our NAIG framework, besides introducing the non-autoregressive generation concept, we also design complexity injection and reasonability refinement, both of which play critical roles in improving data quality and are not straightforward extensions of existing methods.
>
> Therefore, although NAIG is implemented as a practical pipeline, the underlying generation paradigm and the corresponding new modules represent a meaningful step beyond previous data-synthesis approaches.
>
> [1] (Wang et al. NAACL2025) ToolFlow: Boosting LLM Tool-Calling Through Natural and Coherent Dialogue Synthesis
>
> [2] (Yin et al. ACL2025) MAGNET: Multi-turn Tool-use Data Synthesis and Distillation via Graph Translation
>
> [3] (Xu et al. 2025) TOUCAN: Synthesizing 1.5M Tool-Agentic Data from Real-World MCP Environments
>
> [4] (Liu et al. ICLR2025) ToolACE: Winning the Points of LLM Function Calling

---

> ### Author Response · Authors · 2025-11-19
> **Response Part 4**
>
> **Q4: How do you ensure that sub-task sequences in each dialogue reflect realistic correlations rather than random concatenations? Did you compare your generated task‐structure distribution with that of real agent dialogues?**
>
> **A:** As we noted in our response to W4, two components, i.e., high-quality hand-crafted examples and realistic complexity injection, are done to ensure that the sub-task sequences and trajectories reflect meaningful correlations rather than random concatenations. The hand-crafted examples provide realistic structural priors, and the complexity-injection stage further adjusts and refines trajectories to better match real user–agent interaction patterns.
>
> Due to time constraints, we were unable to conduct a rigorous human evaluation comparing the distribution of our generated data with real agent dialogues. However, the substantial performance improvements on benchmarks that closely simulate real-world scenarios, such as $\tau$-Bench, provide indirect but strong evidence that the generated task structures are realistic and effective, rather than arbitrary.
>
>
> **Q5: Are there template or skeleton reuse effects in the initialization phase? How many skeleton variants were used, and how did you guard against structure homogenization?**
>
> **A:** As we point out in W1 and W4, 10 high-quality hand-craft examples are first used for intialization reference. And the example pool will be enriched during generation, with those samples passing offline verification. To ensure diversity, both tool diversity and example diversity (different combination of tool candidates and examples) play critical roles.

---

> > ### Author Response · Authors · 2025-11-26
> >
> > Dear Reviewer,
> >
> > We kindly invite you to review our response and, if you feel your concerns have been adequately addressed, to consider updating your score. If you have any remaining questions, we would be very happy to provide further clarification. Thank you sincerely for your time and thoughtful evaluation!

---

### Official Review · Reviewer_TB5g · 2025-10-31

**Soundness:** 3
**Presentation:** 4
**Contribution:** 3
**Rating:** 4
**Confidence:** 4

**Summary:**

This paper addresses the challenge of generating high-quality multi-turn, multi-step dialogues for agentic task-solving with LLMs, where existing simulation-based methods rely on costly autoregressive interactions and often limit task performance. To tackle this, the authors propose NAIG (Non-Autoregressive Iterative Generation), a three-stage framework that constructs complete conversational trajectories through coarse-grained initialization, iterative refinement via mask-and-fill operations, and offline verification to ensure semantic coherence, contextual consistency, and correctness. Rich experiments were conducted to prove the method's effectivenes.

**Strengths:**

1. This work identifies three important challenges in simulation-based data generation for agentic multi-turn interactions.
2. The dataset has the potential to significantly contribute to research on agentic LLMs, provided it can be publicly released.
3. The paper is clearly written, and the methodology is described in sufficient detail.

**Weaknesses:**

1. The multi-agent simulation process does not incorporate offline verification, and such verification is not an inherent requirement for non-autoregressive methods. This raises concerns about the fairness of comparing NAGI with methods that include offline verification.
2. The improvement of NAIG over Multi-Agent Simulation appears to be marginal. Considering Weakness 1, the core component of the non-autoregressive framework (i.e., NAIG without offline verification) does not seem to provide a meaningful improvement.
3. The reported computational cost may be misleading, as it appears to consider only API calls without accounting for the tokens required for model training and inference. In particular, NAGI likely requires more tokens for both inputs and outputs per API call, which could further increase the actual computational cost.
4. The conclusion that the NAIG generation paradigm supports more effective overall task planning may not be fully supported by the assistant turn counts. Notably, the maximum number of assistant turns in the NAIG training data is higher than in the MAS training data, and the analysis is limited to models trained on τ-Bench. Additional analyses would help strengthen this claim.

**Questions:**

1. The authors mention three key issues in back-and-forth interaction generation. If a single LLM generates the entire interaction via a well-designed prompt and then is refined interactively, would these issues persist, or would new ones emerge?
2. The abbreviation "MAS" is commonly used to refer to multi-agent systems. To avoid potential confusion, I suggest using a different short name for multi-agent simulations.
3. In Figure 4, why is the maximum number of assistant turns in the NAIG training data larger than that in the MAS training data? Could the authors clarify this observation?

---

> ### Author Response · Authors · 2025-11-19
> **Response Part 1**
>
> **W1: The multi-agent simulation process does not incorporate offline verification, and such verification is not an inherent requirement for non-autoregressive methods. This raises concerns about the fairness of comparing NAGI with methods that include offline verification.**
>
> **A:** Thank you for your valuable feedback. Indeed, offline verification plays a critical role in data generation. We would like to clarify that **we have already ensured the inclusion of the same offline verification process for both NAIG and MAS**, as detailed in Lines 288-289 of the paper. Additionally, we control for fairness by using the same candidate tool pool and tool sampling method (random sampling) across both approaches. This ensures that the comparison between NAIG and MAS is conducted under consistent conditions.
>
>
> **W2: The improvement of NAIG over Multi-Agent Simulation appears to be marginal. Considering Weakness 1, the core component of the non-autoregressive framework (i.e., NAIG without offline verification) does not seem to provide a meaningful improvement.**
>
> **A:** We would like to emphasize that the improvement observed with NAIG over MAS is **far from marginal**. For example:
> - For BFCL MT, we observed a significant improvement from 31.38 to 40.25.
> - For ACEBench PA, performance increased from 15.0 to 34.0.
> - On $\tau$-Bench, the overall improvement was from 15.9 to 20.6.
>
> Moreover, as we mentioned in our response to W1, the comparison between MAS and NAIG was conducted under controlled conditions, with the same tool candidates and offline verification applied to both methods. Given these substantial improvements and the fair experimental setup, we believe it is not accurate to characterize the improvement as marginal.
>
>
> **W3: The reported computational cost may be misleading, as it appears to consider only API calls without accounting for the tokens required for model training and inference.**
>
> **A:** Thank you for raising this concern. To address it, we have calculated the token consumption (separating input and output tokens) and the corresponding cost for both methods when generating the final 8K valid samples. Below are the statistics:
>
> | Method | Input Tokens (#) | Output Tokens (#) | Total Cost (GPT-4o Pricing, USD) |
> |:--|:--:|:--:|:--:|
> | MAS |652,145k | 10,712k | 1737 |
> | NAIG | 478,115k | 18,480k | 1380 |
>
> As shown, NAIG requires more output tokens, but its total cost is lower. This is calculated using OpenAI's official pricing, where input tokens cost `$`2.5 per million and output tokens cost `$`10 per million. The primary advantage of NAIG lies in its higher pass rate for offline verification (72.3% vs. 61.1% for MAS, as shown in Table 4), meaning it needs fewer raw samples to achieve the same number of valid final samples. Additionally, NAIG demonstrates similar or even better input token consumption per dialogue, as it requires fewer iterations to produce longer, multi-turn dialogues.
>
> **W4: The conclusion that the NAIG generation paradigm supports more effective overall task planning may not be fully supported by the assistant turn counts. Notably, the maximum number of assistant turns in the NAIG training data is higher than in the MAS training data, and the analysis is limited to models trained on $\tau$-Bench. Additional analyses would help strengthen this claim.**
>
> **A:** Thank you for the opportunity to clarify this point. We define task completion efficiency as how effectively the assistant completes a user task, with fewer turns (whether fewer user interactions or fewer tool calls) clearly indicating better efficiency. Our analysis was conducted on models trained on two constructed datasets (one with MAS and one with NAIG), and tested on $\tau$-Bench. While we agree that a lower average number of assistant turns in the training dataset alone is not direct evidence of better efficiency (as the actual tasks differ between the datasets), the performance on the same test sets in $\tau$-Bench should be considered a valid indicator of efficiency. Additionally, we provide statistics from the training sets to highlight potential sources of the observed task completion efficiency.
>
> To further strengthen our claim, we also evaluated performance on another benchmark, ACEBench's Agent category. The results show that for this evaluation set, the average assistant turn per task is 13.27 for NAIG and 14.27 for MAS, with NAIG requiring fewer turns. This observation holds across two distinct benchmarks, reinforcing the conclusion that models trained with NAIG demonstrate better task completion efficiency.

---

> ### Author Response · Authors · 2025-11-19
> **Response Part 2**
>
> **Q1: The authors mention three key issues in back-and-forth interaction generation. If a single LLM generates the entire interaction via a well-designed prompt and then is refined interactively, would these issues persist, or would new ones emerge?**
>
> **A:** First, let's review the three key issues existed in back-and-forth interaction generation introduced in our Introduction. They are **cost, controllability, and consistency ensurance**. "If a single LLM generates the entire interaction via a well-designed prompt and then is refined interactively" — this is exactly what NAIG has done! We use initialization stage to generate the entire interaction (i.e. trajectory) first, and then the whole trajectory is refined in iterative refinement stage.
>
> According to our response to W3, we have proven that NAIG needs less cost than MAS. While for controllability and consistency, our original design motivation is for these two points. Complexity injection after the whole trajectory is initialized supports different operations while these indeed can be controlled by us. And iterative mask-and-fill refinement enables multiple refinement steps to ensure consistency, while back-and-forth interaction cannot further revise previous turn as later turns are built upon fixed history.
>
> To further strengthen our claim on the consistency, we conduct experiments to assess the coherence of the generated dialogues for both methods (MAS and NAIG), following ToolFlow [1] (where two consecutive turns in one dialogue are treated as the premise and hypothesis, and corresponding entailment ratio and semantic similarity are computed. Please refer to [1] for more details). We compute both EnR (the ratio of entailment) and SS (semantic similarity), and the corresponding results are as follow:
>
> | Method | SS | EnR |
> |:--|:---:|:---:|
> | MAS | 65.23 | 43.60 |
> | NAIG | 68.34 | 50.71 |
>
> The above results validate the better coherence of NAIG-generated data.
>
> [1] (Wang et al. 2025) ToolFlow: Boosting LLM Tool-Calling Through Natural and Coherent Dialogue Synthesis
>
>
> **Q2: The abbreviation "MAS" is commonly used to refer to multi-agent systems. To avoid potential confusion, I suggest using a different short name for multi-agent simulations.**
>
> **A:** Thank you for pointing out that. We would consider to use "MASim" in our later version. However, in this reponse, we preserve the usage of "MAS" for consistency.
>
> **Q3: In Figure 4, why is the maximum number of assistant turns in the NAIG training data larger than that in the MAS training data? Could the authors clarify this observation?**
>
> **A:** Thank you for your insightful question. In our experimental setting, the number of subtasks is sampled from the range $[2,5]$, and each subtask contains between $[1,6]$ steps (Lines 290-291). This leads to the possibility that some resulting dialogues may exceed 30 turns. However, these longer dialogues are typically more complex and are more likely to contain problematic components, which makes them more likely to be filtered out during offline verification. As a result, we observe a peak in the NAIG training data, as shown in Figure 4, reflecting the presence of these longer but more complex task samples.

---

> ### Comment · Reviewer_TB5g · 2025-11-20
> **Response to the rebuttal and score-raising**
>
> Thank you for the authors' response. **Most of my concerns have been addressed**. I believe that "this paper is marginally above the acceptance threshold, but I would not object to its rejection," especially if results for W3 can be included and some statements related to W4 refined for clarity. While the paper presents some useful methods that could aid future work in this area, I find its overall contribution to be somewhat limited (I can give 8/10 if a work investigates how to better leverage the current datasets in terms of data selection or algorithm). Meanwhile, **the experiments are rich, and the generated dataset can be further used**. Therefore, the highest score I can give is 6, and I would like to raise my score from 4 to 6.

---

> > ### Author Response · Authors · 2025-11-20
> >
> > Thank you very much for your thoughtful follow-up and for raising your score! We sincerely appreciate your efforts for reviewing our work. Your suggestions regarding W3 and W4 are well taken. We will incorporate the corresponding results and clarify the statements in the final version.
> >
> > We would also like to briefly highlight what we believe to be the key contributions of our work:
> > - A new non-autoregressive paradigm for multi-turn tool-calling data generation, offering a **fundamentally different** alternative to back-and-forth multi-agent simulation;
> > - Two novel modules (**complexity injection and reasonability refinement**) that significantly enhance the realism, coherence, and diversity of generated trajectories;
> > - Consistent improvements across multiple challenging benchmarks, demonstrating the effectiveness of the generated data and the practical utility of the method.
> >
> > Thank you again for your constructive feedback and for recognizing the potential usefulness of our approach and dataset.

---

### Official Review · Reviewer_HKC7 · 2025-11-02

**Soundness:** 3
**Presentation:** 3
**Contribution:** 3
**Rating:** 4
**Confidence:** 3

**Summary:**

This paper introduces NAIG (Non-Autoregressive Iterative Generation), a novel framework for generating multi-turn dialogues involving function-calling and agentic behavior in large language models (LLMs). Inspired by Non-Autoregressive Translation (NAT) and masked diffusion language models, NAIG generates full conversational trajectories through a non-autoregressive pipeline consisting of three stages. First, a coarse-grained initialization step produces a structurally complete but semantically shallow dialogue skeleton. Second, an iterative refinement stage enhances complexity and semantic coherence using a mask-and-fill strategy inspired by masked language models. Third, an offline verification step applies a combination of rule-based and model-based checks to filter out inconsistent or hallucinated samples. The authors evaluate NAIG on several benchmarks, demonstrating significant improvements over multi-agent simulation (MAS) approaches in multi-turn function-calling accuracy, data generation efficiency, and model generalizability.

**Strengths:**

1. NAIG produces more coherent, context-aware, and complex multi-turn dialogues by combination of structured initialization and iterative refinement.
2. Their method reduces reliance on expensive multi-agent autoregressive interactions, lowering API costs
3. Offline verification enhances data reliability by catching structural and semantic inconsistencies that are hard to detect during generation.
4. Models trained on NAIG-generated data significantly outperform both MAS-based on multi-turn benchmarks.

**Weaknesses:**

1. NAIG generation quality significantly drops when using smaller models (e.g., GPT-4o-mini), limiting its applicability in low-resource settings.
2. Models trained on NAIG-generated data may over-optimize for multi-turn planning, potentially at the expense of performance on single-turn or one-shot queries.
3. The experiments only compare NAIG against a single model despite the diversity of baselines available for tool-calling dialogue generation, raising concerns about the sufficiency and generalizability of the evaluation. \

**Questions:**

1. Please provide detailed statistics of the dataset, such as the number of tools, average number of turns per dialogue, and the average number of tool-calling events per dialogue.
2. Is NAIG extendable to multimodal agentic tasks, such as those involving visual inputs?
3. It would be helpful to include additional comparative analysis against other state-of-the-art tool-based dialogue generation methods to better contextualize NAIG's performance. (e.g. ToolDial: Multi-turn Dialogue Generation Method for Tool-Augmented Language Models, ICLR 2025)

---

> ### Author Response · Authors · 2025-11-19
> **Response Part 1**
>
> **W1: NAIG generation quality significantly drops when using smaller models (e.g., GPT-4o-mini), limiting its applicability in low-resource settings.**
>
> **A:** Thank you for raising this important point. While it's true that generation quality does drop with smaller models, we still observe significant improvement using GPT-4o-mini generated data. For example, our final performance on BFCL-v3 is 60.13 (Table 4), compared to 49.57 (LLama3.1-8B without training, Table 1). Offline verification plays a crucial role in filtering out problematic samples. It's also worth noting that for most agentic data generation methods, the quality of the generated data is closely tied to the model used. For instance, we tested data generation with MAS using GPT-4o-mini (due to time constraints, we generated only around 500 samples), and the pass rate after offline verification (i.e. "Quality" in Table 4) was 51.2%, which is also lower than when using GPT-4o (61.1% from Table 4). This is a common challenge for many data generation approaches.
>
> **W2: Models trained on NAIG-generated data may over-optimize for multi-turn planning, potentially at the expense of performance on single-turn or one-shot queries.**
>
> **A:** While the NAIG pipeline is primarily designed for multi-turn scenarios, the generated data still benefits single-turn tasks. In our experimental setup (Lines 290-291), the number of subtasks is sampled from $[1, 6]$, meaning a small portion of tasks are single-turn. This choice reflects real-world scenarios where users don’t always request complex tasks. The results in Table 1 further demonstrate that NAIG maintains or even improves performance on single-turn evaluation sets: the Non-live subset shows an improvement from 84.21 to 84.94 (MAS drops in this subset), and the Live subset improves from 61.08 to 71.52. These findings confirm that NAIG does not over-optimize for multi-turn planning at the expense of single-turn tasks.

---

> ### Author Response · Authors · 2025-11-19
> **Response Part 2**
>
> **W3&Q3: The experiments only compare NAIG against a single model despite the diversity of baselines available for tool-calling dialogue generation, raising concerns about the sufficiency and generalizability of the evaluation. It would be helpful to include additional comparative analysis against other state-of-the-art tool-based dialogue generation methods to better contextualize NAIG's performance.**
>
> **A:** First, we would like to emphasize that the primary focus of this paper is to propose an alternative to MAS (Multi-Agent Simulation), which is central to many existing tool-calling dialogue generation methods. While there are various tricks built on MAS, the core backbone remains MAS itself. As such, we compare NAIG with MAS in our main experiments, ensuring that all other components (such as tool set sampling and offline verification) are controlled to provide a fair comparison. That said, NAIG is compatible with many techniques from other methods, such as Graph-based sampling for tool set sampling in ToolFlow [1] (where we use random sampling for simplicity in this work) and Multi-Stage Validation in APIGen-MT [2].
>
> We agree that comparing our generated data with those from other existing methods would further strengthen our argument. As a result, we have included a comparison in Table 7 of the appendix, where we evaluate models trained with other publicly available datasets. To provide a more comprehensive comparison, we also add more multi-turn datasets and evaluate across three benchmarks. The results are summarized below:
>
> | Models | BFCL ST | BFCL MT | ACEBench MT | ACEBench PA | $\tau$-Retail | $\tau$-Airline |
> |:--|:--:|:--:|:--:|:--:|:--:|:--:|
> | Llama3.1-8B-Inst | 72.7 | 9.3 | 24.0 | 18.3 | 6.1 | 26.0* |
> | + ToolACE (total ~10K, most are ST) | 80.0 | 8.6 | 51.0 | 14.0 | 13.9 | 22.0 |
> | + Toucan (sampled 8K) | 75.2 | 15.0 | 41.0 | 22.6 | 18.3 | 18.0 |
> | + ToolDial (total ~9K) | 70.1 | 12.3 | 30.0 | 11.4 | 10.4 | 20.0 |
> | + MAS | 79.2 | 31.4 | 48.0 | 15.0 | 21.7 | 10.0 |
> | + NAIG | 78.2 | 40.3 | 51.0 | 34.0 | 25.2 | 16.0 |
>
> Here, "ST" refers to "Single-Turn" and "MT" refers to "Multi-Turn." Toucan [3] is a recent large-scale, diverse multi-turn dataset, from which we sampled 8K samples for a fair comparison. ToolACE [4] and ToolDial [5] are other methods for tool-calling multi-turn dialogue generation. Due to time constraints, we were unable to generate new data with their methods for a fully fair comparison. Instead, we leveraged publicly available data, which we assume the authors have worked hard on the data quality.
>
> The above results show that the data generated by NAIG either matches or outperforms the quality of data generated by existing methods, especially for those multi-turn cases (we have pointed out the issue about Airline category in $\tau$-Bench in original paper (Lines 372-377), the results for Airline are just for reference), highlighting the superiority of our NAIG approach.
>
> [1] (Wang et al. 2025) ToolFlow: Boosting LLM Tool-Calling Through Natural and Coherent Dialogue Synthesis
>
> [2] (Prabhakar et al. 2025) APIGen-MT: Agentic PIpeline for Multi-Turn Data Generation via Simulated Agent-Human Interplay
>
> [3] (Xu et al. 2025) TOUCAN: Synthesizing 1.5M Tool-Agentic Data from Real-World MCP Environments
>
> [4] (Liu et al. 2025) ToolACE: Winning the Points of LLM Function Calling
>
> [5] (Shim et al. 2025) ToolDial: Multi-turn Dialogue Generation Method for Tool-Augmented Language Models
>
>
> **Q1: Please provide detailed statistics of the dataset, such as the number of tools, average number of turns per dialogue, and the average number of tool-calling events per dialogue.**
>
> **A:** Thank you for highlighting the need for these statistics. Below are the detailed statistics for the two generated datasets:
>
>
> | Method | Avg tool # per dialogue | Avg turn # per dialogue | Avg user turn # per dialogue |Avg call turn # per dialogue| Avg call # per user turn |
> |:--|:--:|:--:|:--:|:--:|:--:|
> | MAS | 8.8 | 28.0 | 5.8 | 8.7 | 2.3 |
> | NAIG | 8.4 | 23.7 | 3.4 | 8.5 | 3.7 |
>
>
> As shown in the table, a notable difference between NAIG and MAS is the lower average number of user turns and more call numbers per user turn in NAIG, which suggests improved efficiency in task completion and better focus on tool calling.
>
>
> **Q2: Is NAIG extendable to multimodal agentic tasks, such as those involving visual inputs?**
>
> **A:** While we have not yet extended our method to multimodal scenarios, NAIG is certainly adaptable to such tasks. In multimodal settings, visual inputs (as part of the overall input) would first need to be collected. Then, the corresponding tasks, described in text, along with their trajectories, can be constructed using the NAIG framework.

---

> > ### Comment · Reviewer_HKC7 · 2025-11-25
> >
> > Thank you for the authors' response. I raised the score from 4 to 6.

---

> > > ### Author Response · Authors · 2025-11-26
> > >
> > > Thank you very much for reconsidering your score! We sincerely appreciate your efforts for reviewing our work!

---

### Author Response · Authors · 2025-12-01
**Summary of Rebuttal**

We sincerely thank the reviewers for their insightful and constructive comments, which have greatly improved the quality of our work. We also thank the AC for overseeing the discussion. We have revised the paper accordingly, and summarize the key updates and the concerns they address:

- We introduce a new section with comprehensive **data statistics and quality comparisons**, including a new Table 1 (Lines 293–312). This directly addresses the concerns regarding the lack of dataset analysis and quality comparison (Q1 of Reviewer HKC7, Q1 of Reviewer TB5g, W1&Q1 of Reviewer 9xQr).

- We provide **input/output token counts and total generation cost analysis** (Lines 401–418), addressing the common request across reviewers for evidence of superior efficiency (W3 of Reviewer TB5g, W2&Q2 of Reviewer 9xQr, W1&Q1 of Reviewer JVVr).

- We refine and expand the **comparisons with existing methods** under identical training settings across all three benchmarks (Lines 744–766), resolving the concerns from Reviewer HKC7 (W3&Q3) about lack of evaluation generalizability.

- We include a **failure-case study** to clarify the types of issues encountered during Offline Verification (Lines 775–789), addressing Reviewer 9xQr’s concerns (W3&Q3).

- We refine a few statements for greater clarity (Line 427, Lines 429–431).

With these revisions, we believe we have addressed the primary concerns shared across reviewers, providing clearer evidence of the method’s advantages in cost efficiency, data quality, and comparative strength against existing approaches. We are pleased that two reviewers have already indicated increased scores following the discussion.

Thank you again to the reviewers and the AC for your time and thoughtful evaluation.

---

### Meta-Review · Area_Chair_YPxJ · 2026-01-07

**Summary:**

The paper proposes a non-autoregressive framework, NAIG, for generating multi-turn, agentic dialogues with function calls. Using an approach that performs coarse initialization, iterative mask-and-fill refinement, and offline verification, this approach produces coherent and correct dialogue trajectories efficiently. Experiments show it outperforms the traditional multi-agent simulation methods in terms of accuracy and generalization, while reducing the cost of synthesis.

**Reviewer Concerns:**

Most reviewer concerns seem to be answered during the rebuttals, with well-organized responses. Reviewer TB5g still had concerns: "results for W3 can be included and some statements related to W4 refined for clarity. While the paper presents some useful methods that could aid future work in this area, I find its overall contribution to be somewhat limited." The authors briefly responded to these final concerns.

**Reviewer Scores:**

Tb5g and HKC7 raised their scores after the rebuttals, based on the answers. I feel the other reviewers would keep their scores.

---

### Decision · Program_Chairs · 2026-01-26

Accept (Poster)